# Post-abortion contraceptive use among currently married women in India: New evidence from National Family Health Survey 2019–2021 (NFHS-5)

Joemet Jose[1], Ajit Kumar Kannaujiya[2], Kaushalendra Kumar[3], Lotus McDougal[1], Katherine Hay[1], Abhishek Singh[3*]

1  Center on Gender Equity and Health, University of California San Diego, La Jolla, California, United States of America, 2  Karnataka Health Promotion Trust (KHPT), Bangalore, India, 3  Department of Public Health & Mortality Studies and Centre of Demography of Gender, International Institute for Population Sciences, Mumbai, India

* abhishek@iipsindia.ac.in

## Abstract

Post-abortion contraceptive use is a critical element of reproductive healthcare aimed at preventing unintended pregnancies and promoting reproductive agency. This study investigates changes in post-abortion contraceptive use and factors associated with that use. We use reproductive calendars implemented in 2015–16 and 2019–21 National Family Health Surveys (NFHS) to investigate changes in post-abortion contraceptive use among currently married women age 15–49 in India. We then use 2019–21 NFHS to examine the factors associated with post-abortion contraceptive use. Our analysis is based on a weighted sample of 5,473 women from NFHS-4 and 5,103 women from NFHS-5. The study employs a two-stage estimation procedure using the Inverse Mills Ratio (IMR) framework to address potential biases in abortion reporting. In the second stage, we used a multinomial probit regression model to assess factors influencing post-abortion contraceptive use. Post-abortion contraceptive use increased from 49% in NFHS-4 to 57% in NFHS-5. Multinomial probit regression analysis revealed that gestational age of abortion was negatively associated with post-abortion contraceptive use, while factors such as having a son or prior contraceptive use increased the likelihood. Women who had abortions in private or non-health facilities were less likely to use post-abortion Long-Acting Reversible Contraceptives (LARC), compared to public health facilities. Those who reported unplanned pregnancy or contraceptive failure as the reason for abortion were more likely to use traditional methods of post-abortion contraception. Our findings highlight the importance of integrating family planning services into abortion care and ensuring comprehensive information and counselling on contraceptive options during the post-abortion period, as crucial measures to improve women's health.

**Data availability statement:** The dataset is available on https://doi.org/10.6084/m9.figshare.28658501

**Funding:** This work was supported by the Bill and Melinda Gates Foundation under grant numbers INV-008648 and INV-047355. There was no additional external funding received for this study.

**Competing interests:** The authors have declared that no competing interests exist.

## Introduction

Abortion is a safe health care intervention in settings where abortion is permitted by law and conducted in accordance with recommended guidelines [1]. Global estimates suggest that 55.7 million abortions occurred each year between 2010–14 [2]. A huge majority (88%) of these abortions occurred in developing countries. India is not an exception, with an estimated 15.6 million abortions in 2015 [3]. Global estimates reveal that 45% of the total abortions were unsafe based on the WHO definition of unsafe abortion [2], and this proportion is even higher in India, where 67% of abortions between 2007–2011 were classified as unsafe [4]. When it comes to medical versus surgical abortions, a majority of abortions in India used medical methods of abortion and occurred in settings other than health facilities [5].

While abortions can be life-saving and are considered an essential health care service by the WHO [1], they also carry risk of potential adverse consequences for the health trajectories of both women and children from subsequent pregnancies. Studies have noted that women with a history of abortion are more likely to face recurrent abortions and have an increased risk of spontaneous preterm birth and perinatal death in subsequent pregnancies [6–8]. In low- and middle-income countries including India, abortions are a notable cause of morbidity and mortality among women [4,9–12]. Recent estimates from the UNFPA indicate unsafe abortions as the third leading cause of maternal mortality in India [4]. Where legal, structural, financial, or normative barriers inhibit access to safe abortions, health risks are compounded [12]. Psychological repercussions can also be profound, with some women who undergo abortions grappling with emotions such as guilt, depression, and anxiety [8,13–15]. While medical abortions are safer than other less safe methods, it remains unsafe when provided by informal-sector providers, such as pharmacists, chemists, etc., due to non-compliance with medical standards [5,16]. A majority of users of medical abortions in India purchase the medication from chemists or other informal-sector providers [5,16]. In addition, users receive limited or inaccurate information and counselling [5,16–20].

Post-abortion contraceptive use is crucial for preventing unintended pregnancies and reducing repeated abortions. In the absence of contraceptive methods, women's fertility can return as early as 8–10 days following an abortion [21–23]. Additionally, women may experience adverse health consequences, such as incidence of maternal anaemia, premature rupture of membranes, low birth weight, and preterm delivery in the next pregnancy, if they conceive within six months of abortion [24]. The effective use of post-abortion contraception can play a crucial role in reducing unintended pregnancies and associated maternal and child morbidities and mortality [21,25–27]. Moreover, post-abortion family planning has been identified as a high-impact practice in family planning [28,29].

The use of contraception following an abortion remains limited in India. In 2005–06, 70% of women reported not using any contraceptive method within two months after an abortion [30]. In 2015–16, 64% and 51% did not use any contraceptive methods within the first three and 12 months, respectively, following an abortion [31] and in 2019–21, 43% did not use contraception within 12 months of abortion [32]. While the

WHO recommends a minimum six-month pregnancy interval after miscarriage before attempting another pregnancy [25], many physicians suggest waiting a minimum of only three months before conceiving another child [33]. Despite efforts to expand training and service delivery around comprehensive abortion care in India that includes contraceptive counselling and the launch of universal PPIUCD services across all states in 2010, the increase in post-abortion contraceptive use is low [34].

Numerous studies have examined post-abortion contraceptive use in India, but they have limitations, including issues of small sample sizes and geographical coverage [3,6,20,35,36]. There are only a few nationally representative studies that have looked at the post-abortion contraceptive use in India [30–32], and there are limitations in each. Since NFHS-3 does not directly provide information on abortion, studies that use these data [30] risk conflating estimates of miscarriages, stillbirths and abortions, thus biasing estimates of post-abortion contraceptive use. Prior research in this area has often grouped different contraceptive methods into unclear groups, such as combining IUCD with permanent methods (predominantly female sterilization in India), and combining short acting reversible contraception (SARC) with long-acting reversible contraceptive (LARC), compromising the interpretability of the findings [30–32]. In addition, recent research study did not clearly specify the time frame for initiating contraception post abortion, did not consider the impact of the COVID-19 lockdown, excluded traditional methods of contraception, and did not include, or narrowly defined, important contextual variables such as the use of contraception before abortion, timing of abortion and initiation of contraception after abortion [32].

Most importantly, none of the Indian studies that have researched post-abortion contraceptive use [30–32] have adequately addressed the issues of bias in the reporting of abortion in Indian surveys such as NFHS, DLHS, and IHDS. It is important to acknowledge the sensitivity associated with reporting of abortion in India, as several factors may lead to underreporting or misclassification of abortions. Women may refrain from reporting abortions due to the sensitivity of the topic, including concerns related to sex-selection practices. Mixed methods data collection in one state of India from 2000-2002 found an abortion rate of roughly five times higher than the most recent DHS (NFHS-2, collected in 1998–99) [37]. Moreover, in cases where women do report pregnancy terminations, they may classify them as miscarriages rather than abortions due to legal, cultural and societal stigmas surrounding abortion [3]. In India, the Medical Termination of Pregnancy Act 1971 and 2002 and 2003 amendments enable women to exercise their reproductive right to abortion, but require sanction by a medical provider. So medical abortion directly via a pharmacist or chemist, while common due to privacy, access, ease, normative, and other reasons, is not legal.

Having identified key gaps in the existing research, we investigate the change in post-abortion contraceptive use among currently married women age 15–49 between 2015–16 and 2019–21 after addressing the reporting bias noted earlier. We employed a two-stage estimation procedure using the Inverse Mills Ratio (IMR), proposed by James Heckman [38,39], to address the potential reporting bias in the abortion data. We then examine the factors associated with post-abortion contraceptive use in 2019–21. We accounted for the COVID-19 related lockdowns while examining the associations. The findings of this research may contribute to the development of program goals that emphasize universal access to and provision of family planning services post abortion. Findings of this study may also help in improving reproductive health outcomes for women and enhance the overall well-being of women and children in India.

## Methods

### Data

We used reproductive calendar data drawn from the two most recent rounds of the National Family Health Survey (NFHS), conducted during 2015–16 (NFHS-4) and 2019–21 (NFHS-5) to examine the change in post-abortion contraceptive use in India. The remaining analyses were carried out using reproductive calendar data drawn from NFHS-5. The reproductive calendar in the NFHS-4 and NFHS-5 provides month-by-month information on key reproductive events, such as pregnancies, births, abortions, miscarriages, stillbirths, terminations, contraceptive use, type of contraceptive methods, and reasons

for discontinuation of contraceptive methods for each women over the past 60 months preceding the survey. NFHS is a large-scale multi-round nationally representative cross-sectional household survey conducted by the International Institute for Population Sciences (IIPS) under the stewardship of the Ministry of Health and Family Welfare, Government of India. The NFHS-4 and NFHS-5 collected information on key demographic and reproductive and child health indicators providing national-, state- and district- level estimates.

## Study sample

NFHS used a stratified two-stage sampling design to collect data on key indicators from households and women. Detailed sampling procedure adopted in NFHS can be found in the NFHS reports [40,41]. NFHS-4 and NFHS-5 gathered information from 699,686 and 724,115 women age 15–49, respectively, with response rates of 97%.

Of the women interviewed in NFHS-5, we excluded 537,267 women who did not have any pregnancies in the last five years preceding the survey, leaving 186,848 women who reported 256,939 pregnancy outcomes during this period. Among these pregnancies, 26,722 were non-live births. From this group, we selected 7,696 women who reported abortion from their most recent non-live birth. Since our study focused on the one-year period after abortion, we further excluded 2,836 women who had not yet completed this period, resulting in a sample of 4,860 women. Additionally, we excluded 85 women who had less than four months of calendar data before the pregnancy that resulted in an abortion in order to understand the contraceptive use before abortion. Finally, we excluded 157 women who had missing covariate information. Thus, our final sample included 4,618 currently married women who had completed the one-year period after an abortion. Fig 1 provides the details on the sample selection process. Following a similar process, we selected 5,228 women from NFHS-4. The final weighted sample sizes were 5,103 and 5,473 women in NFHS-5 and NFHS-4, respectively.

## Ethics

NFHS adheres to all ethical guidelines by obtaining informed consent from all respondents before data collection and ensuring their voluntary participation and confidentiality of responses. Furthermore, the data used in this study does not contain any information that may be used to identify the survey respondents. The data is also publicly accessible and can be downloaded from the DHS Program website (dhsprogram.com/).

## Outcome variable

The primary outcome variable of this study is post-abortion contraceptive use. For examining change between 2015–16 and 2019–21, we recoded our primary outcome variable into seven categories: not using any method, sterilization, oral pills, IUD, condom, other modern methods (injectables, diaphragm, foam and jelly, other modern methods), and traditional methods (including rhythm method and withdrawal).

For examining the factors associated with post-abortion contraceptive use, we recoded our primary outcome variable into five categories: not using any method, long acting reversible contraception (LARC), short acting reversible contraception (SARC), sterilization, and traditional methods. LARC refers to long-term and easily reversible contraceptive methods. In this study, LARC includes intrauterine device (IUD) and birth control implants. As no birth control implants were reported in the survey, LARC here consists solely of IUD. SARC refers to contraceptive methods that require regular use or application and are easily reversable. In this study, SARC includes male condom, female condom, lactational amenorrhea, foam and jelly, injectables, oral pills and diaphragm. Traditional methods include withdrawal, rhythm and other traditional methods. We dropped four women who reported using other modern methods post-abortion as we were not sure in which category to include these women.

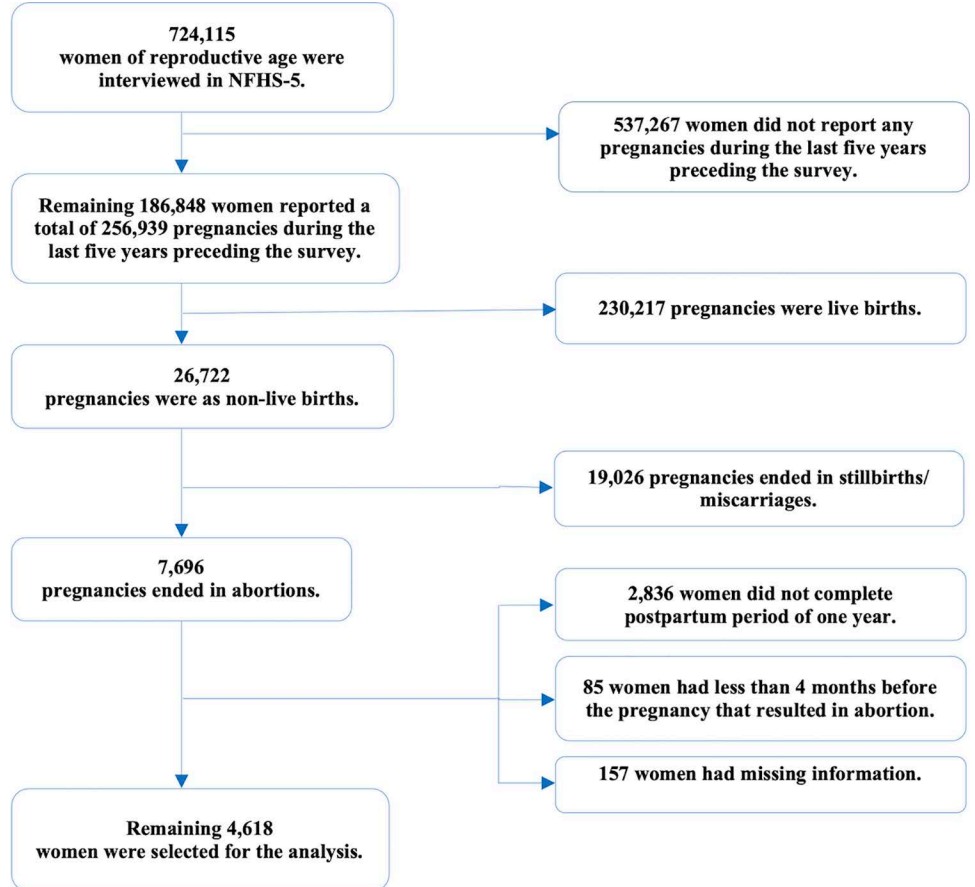

**Fig 1. Sample selection process for currently married women who had completed the one-year period after an abortion using data from NFHS-5, 2019-21.**

### Independent variables

The analysis includes a comprehensive set of independent variables. The abortion-related variables include gestational age of abortion (classified into first month of pregnancy, second month of pregnancy, third month of pregnancy, fourth month of pregnancy or later), the place of the abortion (classified into a public health facility, private health facility, or outside a health facility), person performing abortion (classified into doctor, nurse/ANM/LHV, self, others), and reason for abortion (classified into economic reasons/husband or mother-in-law opposed, unplanned pregnancy/contraceptive failure, complication in pregnancy/foetus had congenital abnormality, health did not permit/last child too young, female foetus/male foetus, others). Contraceptive use-related variables include most recent use of contraception before abortion, classified into four groups: never used, used LARC, used SARC, and used traditional methods. Self-managed abortion includes abortions that are managed by the pregnant person without consultation from a licensed health provider, such as medical abortions, herbal remedies, etc.

The demographic and socio-economic variables used in the study include sex composition of children before abortion (no son, at least one son), women's age at abortion (15–24 years, 25–29 years, 30–49 years), women's schooling (no schooling, primary, secondary, higher), women's exposure to media (no, yes), household type (nuclear, other), household wealth quintiles (lowest, second, middle, fourth, highest), religion (Hindu, Muslim, others), and caste (Scheduled Castes, Scheduled Tribes, Other Backward Classes, others) were also included in the analysis.

Place of residence is classified into urban and rural. We classified the states into geographical regions as given in NFHS-5 (north, central, east, northeast, west, south). The North region of India consists of the states and union territories (UTs) of Chandigarh, Delhi, Haryana, Himachal Pradesh, Jammu & Kashmir, Ladakh, Punjab, Rajasthan, and Uttarakhand. The Central region includes the states of Chhattisgarh, Madhya Pradesh, and Uttar Pradesh. The East region includes the states of Bihar, Jharkhand, Odisha, and West Bengal. The Northeast region encompasses the states of Arunachal Pradesh, Assam, Manipur, Meghalaya, Mizoram, Nagaland, Sikkim, and Tripura. The West region comprises Dadra & Nagar Haveli and Daman & Diu, Goa, Gujarat, and Maharashtra. Finally, the South region includes the Andaman & Nicobar Islands, Andhra Pradesh, Karnataka, Kerala, Lakshadweep, Puducherry, Tamil Nadu, and Telangana.

The NFHS-5 survey was conducted in two phases, with states and UTs divided into two phases. Fieldwork in the states and UTs included in first phase was completed before the COVID-19 related lockdowns were implemented. Fieldwork in the states and UTs included in second phase was affected by India's main COVID-19 related lockdowns, which occurred from 24 March 2020 to May 31 2020. So, we accounted for the timing of the interview, distinguishing between interviews conducted before and after the COVID-19 lockdown (before lockdown, after lockdown).

## Statistical analysis

To address the potential bias in the reporting of abortions we adopted a two-stage estimation procedure based on the Inverse Mills Ratio (IMR) framework proposed by James Heckman [38,39]. IMR is a correction factor used in regression models to account for selection bias, thus ensuring more reliable estimates when certain outcomes, such as abortion reporting, may be systematically underreported. In the first stage, we employed a probit regression model to estimate the likelihood of abortion. While our dependent variable was abortion, we considered several explanatory variables such as the pregnancy outcome before the abortion, the combination of actual and desired children, children ever born, age at marriage, level of schooling, type of household, wealth quintiles, caste, residence and religion. We estimated the IMR from this model. In the second stage, we employed another probit regression model, with contraceptive use as the dependent variable. In this stage, we introduced the IMR generated from the initial model as an independent variable, along with additional covariates. Our multinomial probit model (Table 3) categorized contraceptive use into five distinct categories: LARC, SARC, other modern methods, traditional methods, and no methods.

In the second stage, we employed another probit regression model, with contraceptive use as the dependent variable and the IMR generated from the initial model as an independent variable, along with additional covariates. Our multinomial probit model (Table 3) categorized contraceptive use into five distinct categories: LARC, SARC, other modern methods, traditional methods, and no methods.

The results of our analysis are presented in terms of percentages and adjusted coefficients, each accompanied by 95 percent confidence intervals (CIs). All statistical analyses were conducted using STATA version 17.0.

## Results

Overall post-abortion contraceptive use increased from 49% in NFHS-4–57% in NFHS-5 (Fig 2). There was an increase across most methods, including condom use (from 13% in NFHS-4–16% in NFHS-5) and traditional methods (from 12% in NFHS-4–15% in NFHS-5).

Post-abortion contraceptive use by urban-rural residence in NFHS-4 and NFHS-5 are shown in Fig 3. In NFHS-4, 51% and 48% of women used contraception post-abortion in urban and rural areas respectively. In NFHS-5, 57% and 56% of women used post-abortion in urban and rural areas respectively. While a higher percentage of rural than urban women used oral pills in both the surveys, more urban than rural women used condom. The urban-rural gap in condom use post-abortion narrowed down in NFHS-5. IUD use was higher in urban than in rural areas in both the surveys. While the use of traditional methods post-abortion was high in rural than urban areas in NFHS-4, the use of traditional methods post-abortion in NFHS-5 was similar for urban and rural areas.

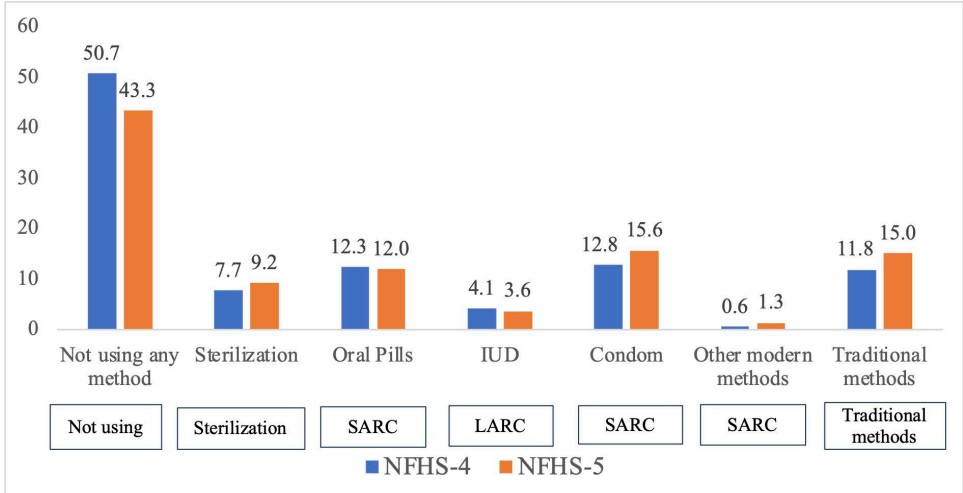

**Fig 2. Percentage distribution of currently married women by different methods of post-abortion contraception, NFHS-4 & NFHS-5.**

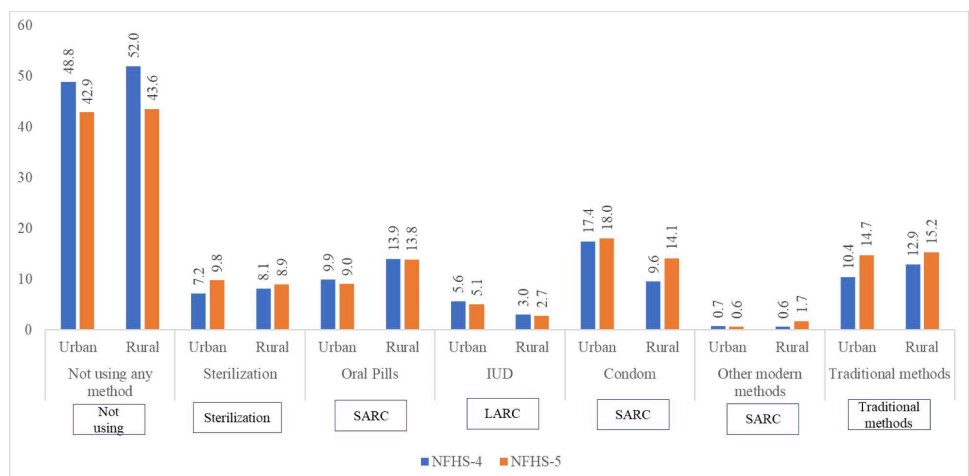

**Fig 3. Percentage of currently married women using various methods of contraception after abortion by urban-rural residence, India, NFHS-4 and NFHS-5.**

One-fifth of women (21%) initiated contraceptive use in month one of abortion. Eight percent, 7%, 8%, and 12% women initiated contraceptive use in month 2, month 3, months 4–5, and month 6 or later, respectively.

**Table 1** provides the distribution of sampled currently married women who had undergone an abortion during the five years preceding NFHS-5. About two-fifths of women had abortion in the second month of pregnancy followed by 26% of women who had abortion in the first month of pregnancy. Fourteen percent of women had abortion in the fourth month or later. A majority of abortions took place in a private health facility (56%). About a quarter of abortions took place outside a health facility (23%). While the majority of abortions (60%) were performed by a doctor, 23% were performed by women themselves. The major reason mentioned for abortion was unplanned pregnancy or contraceptive failure (47%). A little more than three-fifths of women (64%) had not used any contraceptive method before abortion. Twenty percent and 14% of women had used SARC and traditional methods as the method of contraception most recently used before abortion, respectively.

**Table 1. Percentage distribution of currently married women who underwent an abortion and completed one-year post-abortion, India, NFHS-5.**

| Characteristics | Percentage | Number |
|---|---|---|
| Gestational age of abortion | | |
| First month of pregnancy | 26.0 | 1326 |
| Second month of pregnancy | 39.9 | 2035 |
| Third month of pregnancy | 20.0 | 1020 |
| Fourth month of pregnancy or later | 14.1 | 722 |
| Place of abortion | | |
| Public health facility | 21.3 | 1089 |
| Private health facility | 56.1 | 2861 |
| Outside a health facility | 22.6 | 1153 |
| Who performed abortion | | |
| Doctor | 59.7 | 3044 |
| Nurse/ ANM/ LHV | 13.2 | 672 |
| Self | 23.0 | 1180 |
| Others | 4.1 | 207 |
| Reason for abortion | | |
| Economic reasons/husband/mother in law did not want | 6.0 | 308 |
| Unplanned pregnancy/contraceptive failure | 46.5 | 2370 |
| Complication in pregnancy/ foetus had congenital abnormality | 16.2 | 828 |
| Health did not permit/last child too young | 23.2 | 1186 |
| Female foetus/male foetus | 2.2 | 114 |
| Others | 5.9 | 297 |
| Most recent use of contraception before abortion | | |
| Never used | 63.6 | 3244 |
| LARC | 2.1 | 108 |
| SARC | 20.1 | 1024 |
| Traditional methods | 14.2 | 727 |
| Sex composition of children | | |
| No son | 38.3 | 1954 |
| At least one son | 61.7 | 3149 |
| Age at abortion | | |
| 15-24 years | 39.7 | 2025 |
| 25-29 years | 32.6 | 1665 |
| 30-49 years | 27.7 | 1413 |
| Education | | |
| No schooling | 12.6 | 645 |
| Primary | 11.1 | 569 |
| Secondary | 55.6 | 2838 |
| Higher | 20.7 | 1051 |
| Exposure to media | | |
| No | 32.2 | 1641 |
| Yes | 67.8 | 3462 |
| Household type | | |
| Nuclear | 43.1 | 2200 |
| Other | 56.9 | 2903 |
| Wealth quintiles | | |

*(Continued)*

**Table 1.** (Continued)

| Characteristics | Percentage | Number |
|---|---|---|
| Lowest | 13.2 | 671 |
| Second | 18.0 | 920 |
| Middle | 21.5 | 1098 |
| Fourth | 24.2 | 1233 |
| Highest | 23.1 | 1181 |
| Religion | | |
| Hindu | 84.6 | 4319 |
| Muslim | 11.5 | 587 |
| Others | 3.9 | 197 |
| Caste | | |
| Schedule Castes | 23.2 | 1184 |
| Scheduled Tribes | 5.6 | 288 |
| Other Backward Classes | 42.3 | 2155 |
| Others | 28.9 | 1476 |
| Urban-rural residence | | |
| Urban | 37.6 | 1920 |
| Rural | 62.4 | 3183 |
| Region | | |
| North | 10.7 | 545 |
| Central | 22.8 | 1164 |
| East | 24.8 | 1269 |
| Northeast | 5.7 | 290 |
| West | 14.0 | 714 |
| South | 22.0 | 1121 |
| Timing of interview | | |
| Before lockdown | 73.6 | 3755 |
| After lockdown | 26.4 | 1348 |
| **Total** | **100** | **5103** |

About two-fifths of the women were in the age-group 15–24, one-third were in the age-group 25–29, and 28% were in the age-group 30–49 years. Seventy-six percent of women had secondary or higher schooling, 68% had exposure to media, 57% resided in non-nuclear households, and 23% belonged to the highest wealth quintile. A high majority of women (85%) belonged to the Hindu religion. Twenty-three percent, 6%, and 42% women belonged to scheduled castes, scheduled tribes and other backward class respectively. When it comes to urban-rural residence, 62% of women resided in rural areas. A quarter of women resided in the eastern part of India. Twenty-three percent and 22% of women resided in the central and southern parts of India respectively. Only one-fourth of women (26%) were interviewed after the COVID-19 related lockdowns were lifted in India.

Table 2 shows post-abortion contraceptive method use stratified by the independent variables. Initiation of post-abortion contraceptive use decreased as gestational age at abortion increased. For example, 69% of women who had abortion in the first month of pregnancy reported using contraception post-abortion compared with only 37% of women who had abortion in the fourth month or later. Post-abortion contraceptive use was highest among women (69%) who had their abortion outside of a health facility, and lowest among women (51%) who had their abortion in a private health facility. Those women who performed abortion themselves had higher levels of initiation of SARC

**Table 2. Percentage distribution of currently married women who underwent an abortion by type of contraceptive methods, India, NFHS-5.**

| Characteristics | LARC<br>% (95% CI) | SARC<br>% (95% CI) | Sterilization<br>% (95% CI) | Traditional methods<br>% (95% CI) | Overall contra-ception use<br>% (95% CI) |
|---|---|---|---|---|---|
| Gestational age of abortion | | | | | |
| First month of pregnancy | 3.6 (2.4, 5.4) | 36.5 (33.0, 40.2) | 8.9 (6.7, 11.6) | 20.4 (17.5, 23.7) | 69.4 (65.5, 73.0) |
| Second month of pregnancy | 4.5 (3.2, 6.2) | 32.6 (29.6, 35.8) | 10.1 (8.2, 12.4) | 15.0 (13.1, 17.2) | 62.2 (59.1, 65.3) |
| Third month of pregnancy | 3.1 (2.1, 4.7) | 20.9 (17.6, 24.6) | 7.6 (5.4, 10.5) | 11.9 (9.6, 14.7) | 43.5 (39.2, 47.9) |
| Fourth month of pregnancy or later | 1.9 (1.0, 3.5) | 16.8 (13.3, 20.9) | 9.8 (7.1, 13.4) | 8.0 (6.1, 10.5) | 36.5 (32.0, 41.1) |
| Place of abortion | | | | | |
| Public health facility | 6.1 (4.4, 8.3) | 27.4 (24.0, 31.2) | 12.3 (9.7, 15.4) | 11.7 (9.5, 14.2) | 57.5 (53.5, 61.4) |
| Private health facility | 3.2 (2.3, 4.5) | 26.0 (23.7, 28.4) | 9.5 (7.9, 11.4) | 12.6 (11.0, 14.3) | 51.3 (48.6, 54.0) |
| Outside a health facility | 2.2 (1.4, 3.5) | 38.1 (34.3, 41.9) | 5.7 (4.2, 7.7) | 23.3 (20.1, 26.9) | 69.3 (65.6, 72.8) |
| Who performed abortion | | | | | |
| Doctor | 4.1 (3.1, 5.4) | 25.3 (23.1, 27.7) | 10.3 (8.7, 12.1) | 10.8 (9.4, 12.3) | 50.5 (47.9, 53.1) |
| Nurse/ ANM/ LHV | 3.3 (2.1, 5.2) | 29.1 (24.6, 34.1) | 12.0 (8.8, 16.2) | 16.2 (13.1, 19.8) | 60.7 (55.7, 65.4) |
| Self | 2.7 (1.8, 4.1) | 37.0 (33.2, 40.9) | 5.3 (3.9, 7.3) | 24.0 (20.6, 27.7) | 69.0 (65.2, 72.6) |
| Others | 1.9 (0.6, 5.2) | 38.2 (30.1, 47.0) | 6.6 (3.3, 12.7) | 17.9 (12.4, 25.0) | 64.5 (55.2, 72.8) |
| Reason for abortion | | | | | |
| Economic reasons/husband/mother in law did not want | 5.7 (2.9, 10.7) | 36.9 (29.5, 45.0) | 14.7 (9.5, 21.9) | 12.0 (8.3, 17.1) | 69.3 (61.8, 75.9) |
| Unplanned pregnancy/contraceptive failure | 3.9 (2.9, 5.3) | 34.9 (32.2, 37.6) | 11.4 (9.5, 13.5) | 20.5 (18.3, 22.8) | 70.7 (68.0, 73.2) |
| Complication in pregnancy/ foetus had congenital abnormality | 3.0 (1.4, 6.4) | 13.9 (11.3, 17.1) | 6.1 (4.1, 9.1) | 7.6 (5.6, 10.2) | 30.7 (26.5, 35.2) |
| Health did not permit/last child too young | 3.4 (2.2, 5.0) | 26.9 (23.3, 30.8) | 5.3 (3.8, 7.4) | 11.2 (9.1, 13.8) | 46.8 (42.8, 50.9) |
| Female foetus/male foetus | 0.1 (0.0, 1.0) | 37.5 (26.3, 50.3) | 22.8 (13.7, 35.4) | 5.2 (2.0, 12.5) | 65.6 (54.6, 75.2) |
| Others | 2.5 (1.2, 5.2) | 21.9 (15.7, 29.5) | 5.7 (3.0, 10.6) | 10.5 (6.5, 16.7) | 40.6 (32.5, 49.2) |
| Most recent use of contraception before abortion | | | | | |
| Never used | 3.4 (2.5, 4.6) | 20.2 (18.4, 22.3) | 8.2 (6.8, 9.9) | 10.5 (9.1, 12.0) | 42.3 (39.9, 44.8) |
| LARC | 13.2 (6.2, 26.1) | 25.0 (14.5, 39.5) | 5.4 (2.1, 13.1) | 17.3 (8.2, 33.0) | 60.9 (45.5, 74.4) |
| SARC | 2.8 (1.9, 4.2) | 57.3 (52.7, 61.8) | 11.4 (8.7, 14.8) | 9.1 (7.1, 11.7) | 80.7 (76.9, 83.9) |
| Traditional methods | 4.0 (2.5, 6.5) | 29.1 (24.8, 33.6) | 11.3 (8.2, 15.4) | 41.8 (37.2, 46.6) | 86.2 (82.8, 89.1) |
| Sex composition of children | | | | | |
| No son | 2.3 (1.6, 3.4) | 21.3 (18.9, 23.9) | 3.4 (2.2, 5.3) | 8.9 (7.5, 10.5) | 35.9 (33.0, 38.9) |
| At least one son | 4.4 (3.4, 5.7) | 33.8 (31.6, 36.2) | 12.8 (11.2, 14.7) | 18.5 (16.7, 20.4) | 69.6 (67.3, 71.8) |
| Age at abortion | | | | | |
| 15-24 years | 3.9 (2.6, 5.7) | 25.6 (23.0, 28.3) | 6.2 (4.7, 8.1) | 11.9 (10.0, 14.1) | 47.6 (44.5, 50.7) |
| 25-29 years | 2.8 (1.9, 4.0) | 30.9 (27.8, 34.1) | 11.5 (9.4, 13.9) | 13.6 (11.6, 15.8) | 58.7 (55.2, 62.1) |
| 30-49 years | 4.1 (3.0, 5.7) | 31.8 (28.5, 35.3) | 11.0 (8.6, 13.8) | 20.4 (17.7, 23.4) | 67.3 (63.9, 70.6) |
| Education | | | | | |
| No schooling | 1.2 (0.7, 2.2) | 31.0 (26.2, 36.2) | 12.7 (9.7, 16.5) | 21.2 (17.2, 25.7) | 66.1 (61.4, 70.5) |
| Primary | 2.8 (1.5, 5.0) | 32.2 (27.4, 37.5) | 15.9 (12.0, 20.7) | 13.1 (9.9, 17.2) | 63.9 (57.9, 69.6) |
| Secondary | 4.0 (3.1, 5.2) | 28.2 (25.9, 30.6) | 8.8 (7.3, 10.5) | 15.3 (13.6, 17.3) | 56.3 (53.7, 58.9) |
| Higher | 4.4 (2.7, 7.2) | 28.4 (24.6, 32.5) | 4.8 (2.9, 7.7) | 10.5 (8.1, 13.4) | 48.1 (43.6, 52.6) |
| Exposure to media | | | | | |
| No | 2.9 (2.1, 4.1) | 29.7 (26.8, 32.7) | 11.3 (9.3, 13.7) | 16.9 (14.6, 19.4) | 60.8 (57.5, 63.9) |
| Yes | 3.9 (3.0, 5.1) | 28.7 (26.6, 30.9) | 8.3 (6.9, 9.8) | 13.8 (12.3, 15.5) | 54.8 (52.3, 57.2) |
| Household type | | | | | |
| Nuclear | 3.5 (2.4, 5.0) | 31.6 (29.0, 34.4) | 11.3 (9.4, 13.4) | 17.3 (15.1, 19.7) | 63.6 (60.8, 66.4) |

*(Continued)*

Table 2.  (Continued)

| Characteristics | LARC | SARC | Sterilization | Traditional methods | Overall contra-ception use |
|---|---|---|---|---|---|
| | % (95% CI) | % (95% CI) | % (95% CI) | % (95% CI) | % (95% CI) |
| Other | 3.7 (2.9, 4.7) | 27.1 (24.9, 29.4) | 7.7 (6.3, 9.4) | 12.9 (11.4, 14.6) | 51.4 (48.8, 54.0) |
| Wealth quintiles | | | | | |
| Lowest | 1.4 (0.8, 2.4) | 34.7 (30.1, 39.7) | 11.5 (8.8, 14.9) | 17.2 (13.8, 21.3) | 64.8 (59.8, 69.5) |
| Second | 3.4 (2.2, 5.2) | 29.4 (25.7, 33.4) | 13.2 (10.1, 17.0) | 16.6 (13.7, 20.1) | 62.6 (58.6, 66.5) |
| Middle | 3.7 (2.5, 5.6) | 28.3 (24.9, 32.0) | 7.9 (5.8, 10.7) | 14.1 (11.6, 17.0) | 54.0 (50.0, 57.9) |
| Fourth | 3.4 (2.0, 5.6) | 26.0 (22.5, 29.9) | 7.7 (5.8, 10.1) | 13.9 (11.2, 17.2) | 51.0 (46.8, 55.1) |
| Highest | 5.1 (3.4, 7.7) | 29.4 (25.7, 33.3) | 7.7 (5.4, 10.8) | 13.7 (11.3, 16.4) | 55.9 (51.5, 60.2) |
| Religion | | | | | |
| Hindu | 3.9 (3.1, 4.9) | 28.7 (26.9, 30.6) | 9.5 (8.3, 10.9) | 15.2 (13.8, 16.6) | 57.3 (55.2, 59.4) |
| Muslim | 1.3 (0.6, 2.8) | 34.6 (28.9, 40.7) | 7.6 (4.5, 12.5) | 12.5 (8.7, 17.6) | 55.9 (49.6, 62.1) |
| Others | 3.8 (1.9, 7.4) | 19.2 (14.2, 25.5) | 8.0 (3.7, 16.4) | 13.6 (9.6, 19.0) | 44.6 (36.3, 53.3) |
| Caste | | | | | |
| Schedule Castes | 3.0 (1.9, 4.7) | 27.7 (24.3, 31.4) | 10.9 (8.4, 14.1) | 15.1 (7.7, 12.9) | 56.7 (52.4, 60.9) |
| Scheduled Tribes | 2.3 (1.2, 4.4) | 30.1 (24.4, 36.5) | 8.0 (3.7, 16.4) | 17.5 (12.4, 18.2) | 57.9 (50.8, 64.7) |
| Other Backward Classes | 3.3 (2.5, 4.5) | 27.9 (25.3, 30.6) | 7.9 (6.5, 9.6) | 13.4 (13.1, 23.0) | 52.6 (49.7, 55.4) |
| Others | 4.7 (3.1, 7.1) | 31.6 (28.1, 35.4) | 10.0 (0.0, 12.9) | 16.1 (11.8, 15.2) | 62.5 (58.6, 66.1) |
| Urban-rural residence | | | | | |
| Urban | 5.1 (3.6, 7.1) | 27.7 (24.8, 30.9) | 9.8 (7.6, 12.4) | 14.6 (12.3, 17.1) | 57.1 (53.6, 60.6) |
| Rural | 2.7 (2.1, 3.5) | 29.8 (27.8, 31.9) | 8.9 (7.7, 10.3) | 15.0 (13.5, 16.5) | 56.4 (54.2, 58.6) |
| Region | | | | | |
| North | 6.1 (4.1, 8.8) | 32.9 (29.1, 36.9) | 7.9 (5.5, 11.1) | 17.9 (14.5, 21.8) | 64.7 (60.5, 68.7) |
| Central | 3.0 (2.1, 4.5) | 38.5 (34.9, 42.2) | 8.1 (6.3, 10.4) | 18.1 (15.4, 21.2) | 67.8 (64.0, 71.3) |
| East | 1.7 (1.0, 3.0) | 35.8 (31.8, 39.9) | 11.0 (8.7, 13.8) | 20.9 (17.8, 24.4) | 69.4 (65.7, 72.8) |
| Northeast | 4.9 (3.2, 7.6) | 42.2 (37.3, 47.3) | 1.4 (0.7, 3.1) | 27.2 (23.0, 31.8) | 75.8 (71.1, 79.9) |
| West | 5.0 (2.8, 9.0) | 22.1 (17.2, 28.1) | 10.9 (7.3, 16.2) | 6.7 (4.2, 10.3) | 44.7 (37.8, 51.9) |
| South | 3.8 (2.2, 6.6) | 10.7 (8.3, 13.7) | 10.0 (7.6, 13.1) | 5.0 (3.6, 6.9) | 29.5 (25.6, 33.8) |
| Timing of interview | | | | | |
| Before lockdown | 3.5 (2.7, 4.6) | 28.6 (26.6, 30.8) | 9.7 (8.3, 11.3) | 13.6 (12.2, 15.1) | 55.4 (53.1, 57.7) |
| After lockdown | 3.8 (2.7, 5.3) | 30.2 (27.1, 33.4) | 7.9 (6.2, 10.1) | 18.3 (15.8, 21.0) | 60.2 (56.7, 63.6) |
| **Total** | **3.6 (2.9, 4.5)** | **29.1 (27.4, 30.8)** | **9.2 (8.1, 10.5)** | **14.8 (13.6, 16.2)** | **56.7 (54.8, 58.6)** |

Note: % - Percent, CI - Confidence Interval

or traditional methods post-abortion, and the overall contraceptive use after abortion was highest among this group. More women used post-abortion contraception if the reason for abortion was unplanned pregnancy or contraceptive failure. Those women who have used traditional methods of contraception as the most recent method before abortion had highest level of post-abortion contraceptive use. This group was followed by women who had used SARC as the most recent method before abortion. Having a son was associated with higher post-abortion contraceptive use. Seventy percent of women having a son reported post-abortion contraceptive use compared with only 36% of women not having a son.

Use of contraceptives post-abortion increased with increasing age of women. For example, 48% of women age 15–24 reported using a contraceptive method post-abortion compared with 67% of women age 30–49. In contrast, post-abortion

contraceptive use declined with increases in women's schooling, with a prevalence of 66% among women with no schooling and 48% among women with higher schooling. Women who had exposure to media reported lower post-abortion contraceptive use compared with women who had no exposure to media. Post-abortion contraceptive use among women from nuclear households was 12 percentage points higher than women from non-nuclear households. A higher percentage of Hindu women and women who did not belong to scheduled castes, scheduled tribes or other backward class reported post-abortion contraceptive use compared with their counterparts. Post-abortion contraceptive use was highest among women residing in the northeast part of India followed by women residing in the east part. Post-abortion contraceptive use was 5 percentage points higher among women who were interviewed after the COVID-19 related lockdowns were lifted in India compared with women who were interviewed before the lockdown.

SARCs were the most common methods initiated post-abortion among women of all ages. Barring a couple of exceptions, traditional methods were the second most commonly-used contraceptive methods post-abortion among women irrespective of their selected characteristics.

Table 3 presents the results from the multinomial probit model examining association between post-abortion contraceptive use and independent variables. Compared to women who had abortion in the first month of pregnancy, women who had abortion in the fourth month or later were less likely to initiate SARC and traditional methods of contraception than to not initiate contraception. Similarly, women who had an abortion in their third month of pregnancy were less likely to initiate SARC, sterilization or traditional methods of contraception than to not initiate contraception. Women who had abortion in the second month were less likely to initiate traditional methods of contraception than to not initiate contraception. Women who had abortion in a private health facility or outside a health facility were less likely than women who had abortion in a public health facility to use LARC post-abortion. Those who mentioned unplanned pregnancy or contraceptive failure as the reason for abortion were less likely to use LARC (coefficient value of -0.46) post-abortion. Women who had abortion due to complication in pregnancy or foetus had congenital abnormality were less likely to use LARC, SARC or sterilization. Similarly, women who resorted to abortion due to health issues or the child being too young were less likely to use LARC (coefficient value of -0.70), SARC (coefficient value of -0.53) or sterilization (coefficient value of -0.87). Women who mentioned female foetus or male foetus as the reason for abortion were less likely to initiate LARC (coefficient value of -1.68) than to not initiate contraception.

Type of method most recently used before abortion was associated with post-abortion contraceptive. Women who had used LARC as the most recent method before abortion were more likely than women who did not use any contraceptive method before abortion to initiate LARC than to not initiate contraception. Women who had used SARC or traditional methods as the most recent method before abortion were more likely than women who did not use any contraceptive method before abortion to initiate all groups of contraceptive methods than to not initiate contraception. Women having a son were more likely than women not having a son to initiate all the four types of contraceptive methods than to not initiate contraception. While women age 25–29 were more likely to use sterilization post-abortion, women age 30–49 were more likely to use traditional methods compared to women age 15–24.

Women with primary schooling were less likely than women with no schooling to use traditional methods (coefficient value of -0.38) post-abortion. Women with secondary schooling were more likely than women with no schooling to use LARC (coefficient value of 0.58) post-abortion. Women with higher schooling had a lower likelihood of using sterilization (coefficient value of -0.54) post-abortion. Women having exposure to media had higher likelihood of using SARC (coefficient value of 0.22) post-abortion. Women belonging to non-nuclear households were less likely to use SARC or traditional methods (coefficient values of -0.51 and -0.46, respectively) post-abortion. While women living in households in the second-lowest wealth quintile were more likely than women from lowest wealth quintile to use LARC, women from middle, fourth or fifth (highest) wealth quintile were more likely than women in lowest wealth quintiles to use SARC post-abortion. Women from the fourth wealth quintile were less likely than women in first wealth quintile to use sterilization post-abortion. Women from the second, middle, fourth or fifth wealth quintiles were more likely than women in first wealth quintile to use

**Table 3. Multinomial probit model of post-abortion contraceptive methods (LARC, SARC, sterilization and traditional methods), India, NFHS-5.**

| Characteristics | LARC vs. no method | SARC vs. no method | Sterilization vs. no methods | Traditional methods vs. no methods |
|---|---|---|---|---|
| Gestational age of abortion | | | | |
| First month of pregnancy ® | | | | |
| Second month of pregnancy | 0.08 (-0.25, 0.41) | -0.20 (-0.41, 0.02) | -0.11 (-0.40, 0.18) | -0.35* (-0.57, -0.12) |
| Third month of pregnancy | -0.28 (-0.66, 0.09) | -0.54* (-0.79, -0.29) | -0.48* (-0.82, -0.14) | -0.44* (-0.73, -0.16) |
| Fourth month of pregnancy or later | -0.45 (-0.92, 0.01) | -0.56* (-0.85, -0.27) | -0.09 (-0.48, 0.29) | -0.54* (-0.85, -0.23) |
| Place of abortion | | | | |
| Public health facility ® | | | | |
| Private health facility | -0.57* (-0.85, -0.28) | -0.17 (-0.38, 0.05) | -0.21 (-0.46, 0.05) | 0.03 (-0.20, 0.26) |
| Other | -0.74* (-1.21, -0.28) | -0.03 (-0.39, 0.32) | -0.13 (-0.56, 0.31) | 0.15 (-0.23, 0.52) |
| Who performed abortion | | | | |
| Doctor ® | | | | |
| Nurse/ANM/LHV | 0.23 (-0.44, 0.90) | -0.09 (-0.53, 0.35) | 0.46 (-0.12, 1.04) | 0.01 (-0.46, 0.49) |
| Self | 0.17 (-0.53, 0.88) | -0.19 (-0.66, 0.28) | 0.41 (-0.22, 1.03) | 0.04 (-0.46, 0.55) |
| Others | 0.27 (-0.37, 0.91) | -0.17 (-0.58, 0.23) | -0.19 (-0.70, 0.31) | 0.12 (-0.30, 0.53) |
| Reason for abortion | | | | |
| Economic reasons/husband/mother in law did not want ® | | | | |
| Unplanned pregnancy/contraceptive failure | -0.46* (-0.90, -0.03) | -0.26 (-0.59, 0.07) | -0.22 (-0.61, 0.17) | 0.05 (-0.31, 0.40) |
| Complication in pregnancy/foetus had congenital abnormality | -0.84* (-1.39, -0.29) | -0.78* (-1.16, -0.40) | -0.77* (-1.28, -0.27) | -0.26 (-0.70, 0.18) |
| Health did not permit/last child too young | -0.70* (-1.17, -0.24) | -0.53* (-0.89, -0.17) | -0.87* (-1.31, -0.43) | -0.32 (-0.70, 0.06) |
| Female foetus/male foetus | -1.68* (-2.60, -0.77) | 0.35 (-0.22, 0.93) | 0.47 (-0.16, 1.11) | -0.47 (-1.31, 0.36) |
| Others | -0.93* (-1.54, -0.32) | -0.72* (-1.18, -0.25) | -0.70* (-1.29, -0.11) | -0.47 (-0.99, 0.06) |
| Most recent use of contraception before abortion | | | | |
| Never used ® | | | | |
| LARC | 0.96* (0.31, 1.60) | 0.22 (-0.31, 0.75) | 0.12 (-0.59, 0.83) | 0.48 (-0.19, 1.15) |
| SARC | 0.60* (0.30, 0.90) | 1.38* (1.18, 1.58) | 0.92* (0.65, 1.20) | 0.41* (0.17, 0.65) |
| Traditional methods | 0.97* (0.62, 1.32) | 0.94* (0.69, 1.19) | 1.16* (0.81, 1.50) | 1.63* (1.38, 1.88) |
| Sex composition of children | | | | |
| No Son ® | | | | |
| At least one son | 0.82* (0.50, 1.14) | 0.64* (0.46, 0.82) | 1.16* (0.83, 1.50) | 0.66* (0.46, 0.85) |
| Age at abortion | | | | |
| 15-24 years ® | | | | |
| 25-29 years | -0.26 (-0.58, 0.06) | 0.01 (-0.18, 0.21) | 0.28* (0.01, 0.54) | 0.03 (-0.19, 0.25) |
| 30-49 years | -0.04 (-0.40, 0.32) | 0.08 (-0.13, 0.29) | 0.14 (-0.15, 0.43) | 0.33* (0.09, 0.56) |
| Education | | | | |
| No schooling ® | | | | |
| Primary | 0.23 (-0.23, 0.69) | 0.10 (-0.22, 0.42) | 0.17 (-0.18, 0.53) | -0.38* (-0.75, -0.02) |
| Secondary | 0.58* (0.23, 0.94) | 0.18 (-0.10, 0.47) | -0.05 (-0.37, 0.28) | -0.09 (-0.39, 0.21) |
| Higher | 0.44 (0.00, 0.89) | 0.12 (-0.22, 0.47) | -0.54* (-1.00, -0.07) | -0.36 (-0.73, 0.01) |
| Exposure to mass-media | | | | |
| No ® | | | | |
| Yes | 0.04 (-0.25, 0.33) | 0.22* (0.03, 0.41) | -0.08 (-0.33, 0.18) | 0.06 (-0.14, 0.27) |

*(Continued)*

**Table 3.** (Continued)

| Characteristics | LARC vs. no method | SARC vs. no method | Sterilization vs. no methods | Traditional methods vs. no methods |
|---|---|---|---|---|
| Family type | | | | |
| Nuclear ® | | | | |
| Other | -0.25 (-0.53, 0.02) | -0.51* (-0.68, -0.33) | -0.14 (-0.37, 0.10) | -0.46* (-0.67, -0.26) |
| Wealth quintiles | | | | |
| Lowest ® | | | | |
| Second | 0.44* (0.04, 0.85) | 0.17 (-0.11, 0.45) | 0.10 (-0.28, 0.47) | 0.34* (0.02, 0.65) |
| Middle | 0.38 (-0.07, 0.84) | 0.33* (0.03, 0.63) | -0.33 (-0.73, 0.06) | 0.40* (0.06, 0.74) |
| Fourth | 0.18 (-0.32, 0.68) | 0.43* (0.09, 0.77) | -0.57* (-1.03, -0.12) | 0.46* (0.07, 0.85) |
| Highest | 0.45 (-0.12, 1.02) | 0.80* (0.41, 1.18) | -0.42 (-0.98, 0.14) | 0.67* (0.23, 1.10) |
| Religion | | | | |
| Hindu ® | | | | |
| Muslim | -0.96* (-1.41, -0.51) | -0.22 (-0.50, 0.06) | -0.21 (-0.66, 0.24) | -0.52* (-0.85, -0.19) |
| Others | -0.28 (-0.77, 0.22) | -0.40* (-0.80, 0.00) | -0.07 (-0.65, 0.51) | -0.25 (-0.67, 0.17) |
| Caste | | | | |
| Schedule Castes ® | | | | |
| Scheduled Tribes | -0.14 (-0.64, 0.35) | -0.08 (-0.44, 0.28) | 0.00 (-0.64, 0.65) | -0.06 (-0.45, 0.32) |
| Other Backward Class | 0.13 (-0.18, 0.44) | -0.01 (-0.22, 0.20) | -0.12 (-0.40, 0.16) | -0.02 (-0.25, 0.20) |
| Others | 0.51* (0.13, 0.88) | 0.22 (-0.04, 0.47) | 0.07 (-0.26, 0.40) | 0.28* (0.01, 0.55) |
| Urban-rural residence | | | | |
| Urban | | | | |
| Rural | -0.31* (-0.61, 0.00) | 0.02 (-0.18, 0.22) | -0.07 (-0.35, 0.22) | -0.12 (-0.34, 0.10) |
| Region | | | | |
| North ® | | | | |
| Central | -0.25 (-0.65, 0.16) | 0.36* (0.09, 0.63) | -0.27 (-0.64, 0.10) | 0.13 (-0.19, 0.45) |
| East | -0.52* (-0.94, -0.10) | 0.41* (0.12, 0.70) | -0.17 (-0.57, 0.23) | 0.46* (0.13, 0.80) |
| Northeast | 0.44 (-0.18, 1.07) | 1.42* (1.01, 1.83) | -1.94* (-2.64, -1.23) | 1.24* (0.78, 1.70) |
| West | -0.33 (-0.83, 0.16) | -0.30 (-0.65, 0.06) | -0.24 (-0.69, 0.21) | -0.47* (-0.88, -0.06) |
| South | -0.32 (-0.76, 0.12) | -0.36* (-0.69, -0.02) | -0.19 (-0.65, 0.27) | -0.45* (-0.83, -0.07) |
| **Timing of interview** | | | | |
| Before lockdown ® | | | | |
| After lockdown | 0.08 (-0.25, 0.41) | -0.16 (-0.35, 0.04) | -0.04 (-0.29, 0.21) | 0.13 (-0.07, 0.33) |
| Inverse Mills Ratio (IMR) | 4.67* (0.40, 8.94) | 11.70* (9.01, 14.40) | -6.70* (-10.08, -3.32) | 7.22* (4.18, 10.26) |

**Note:** * P-value less than 0.05, CI: Confidence Interval; ® Reference category

traditional methods post-abortion. Compared to Hindus, Muslim women were less likely to use LARC or traditional methods post-abortion.

Rural women were less likely than urban women to use LARC post-abortion. Regional variations were apparent, with women residing in the east region exhibiting lower likelihood of using LARC (coefficient value of -0.52) post-abortion compared to those residing in the north region. However, these women were more likely to use SARC and traditional methods of contraception post-abortion. Women residing in the central region were more likely than women residing in the north region to use SARC post-abortion. Women residing in the northeast region had significantly higher likelihood of using SARC (coefficient value of 1.42) or traditional methods (coefficient value of 1.24) post-abortion compared to women

residing in the north region. Compared to women residing in the north, women residing in the northeast region were less likely to use sterilization (coefficient value of -1.94) post-abortion. Women residing in the west were less likely than women residing in the north to use traditional methods post-abortion. Women residing in the south region were less likely to use SARC or traditional methods (coefficient values of -0.36 and -0.45, respectively) post-abortion compared to women residing in the north region. There was no association between COVID-19 related lockdown and post-abortion contraceptive use.

## Discussion

The findings of our study, adjusted for reporting bias and key covariates, provide valuable insights into post-abortion contraceptive use in India. While post-abortion contraceptive use offers important pathways to improved health outcomes and increased reproductive autonomy, our findings indicate that only 57% of women reported initiating contraceptive use during the 12 months post-abortion in 2019–21. Post-abortion contraceptive use increased by 8 percentage points between 2015–16 and 2019–21. This is less than the overall increase in contraceptive use (57% in 2015–16–69% in 2019–21) during this period [40,41]. Use of condom and traditional methods of contraception made up the bulk of growth in PAFP, both increased by 3 percentage points each during this period. This large increase in post-abortion contraceptive use in India since 2005–06, when post-abortion contraceptive use was estimated at 30% [30] may be at least partially attributed to various government efforts under National Health Mission (NHM), originally launched as National Rural Health Mission (NRHM) in the year 2005, with the aim to bring about architectural corrections in the India's public health system to reach to the poor and the marginalized population sub-groups [42]. Various guidelines, issued under the NHM, focus on post-abortion contraceptive use counselling and time of initiation of contraceptive methods post-abortion with emphasis on oral pills, condoms, injections, IUCD and sterilization [21,34]. However, despite these efforts, the overall gain in post abortion family planning and the mix of methods promoted through these guidelines are not reflected in the relatively modest gains (and method mix) of the latest data on post-abortion family planning. Some policy implications may be gleaned from looking at the factors associated with use.

Several socio-economic and demographic factors were associated with post-abortion contraceptive use. Sex composition of children influenced post-abortion contraceptive use; women with at least one son reported higher contraceptive use post-abortion compared to women with no sons. This finding is in line with a large body of research indicating the manifestation of son preference in contraceptive choice in India [43–46]. Our finding underscores that sons continue to matter while deciding whether to use a contraceptive method post-abortion. Given the important social, economic and religious roles that sons continue to play in the Indian society, sonless women may opt to not use contraception after an abortion to adhere to the normative desire to have a son [46]. Policies and interventions that challenge son preference could be instrumental in increasing post-abortion contraceptive use among women with no sons.

Age of women was associated with post-abortion contraceptive use; older women were more likely than younger women to use a contraceptive method post abortion. Older women were also more likely to use traditional methods of contraception, a finding consistent with prior research [6,30]. Pregnancy-month of abortion was negatively associated with post-abortion contraceptive use. Our finding is consistent with the prior studies; women who had abortion in the second trimester were less likely to use post-abortion contraception compared with women who had abortion in the first trimester [6,30]. These findings have important policy implications. Policies may focus more on younger women, and promoting earlier health seeking within pregnancy, to increase post-abortion contraceptive use in India. Given that older women were more likely to use traditional contraceptives post-abortion than no method, this demographic may benefit from outreach to share information on the lower effectiveness of traditional methods [47]. This is also an area that merits further research, to better understand factors influencing this group's preference for traditional methods after abortion.

Use of contraceptive methods before abortion was associated with higher post-abortion contraceptive use in our study. While women who used LARC as the most recent method before abortion were more likely to use LARC post-abortion, women who used SARC as the most recent method before abortion were more likely to use all the four types of

contraceptive methods post-abortion. This finding underscores the important role that an effective family planning programme can play in reducing unintended pregnancies and improving the health of women and their children, and suggests that engagement with health systems may promote subsequent health service utilization. Women who had abortion in a private health facility were less likely than women who had abortion in a public health facility to use LARC. This finding is consistent with the finding of Gaur et al. [30] and Banerjee et al. [6] which reported higher use of IUCD or permanent methods among women who had abortion in a public facility compared with a private facility. This finding is plausible given that the Government of India provides incentives for IUCD insertions and that public health facilities are more aligned and adhere to the government guidelines issued from time to time [48]. In 2019–21, 13% of women using PPIUCD received compensation. In India, the availability of contraceptive methods in private facilities providing abortion-related services are often more limited than in public facilities providing such services [20]. Our analysis further suggests that over half (56%) of women had abortions in a private health facility. Our findings have implications for the Indian family planning programme, which has a strong focus on intrauterine devices [21]. They also suggest the importance of an expanded basket of method availability within all facilities that provide abortions.

Women exposed to media were more likely to use SARC in our study. Exposure to mass- media was also associated with use of modern temporary methods (includes both LARC and SARC) in the study by Zavier and Padmadas [30]. This finding indicates that media can play an important role in increasing post-abortion contraceptive use in India.

Regional disparities in post-abortion contraceptive use also warrant attention, with variations in post-abortion contraceptive use observed across different geographical regions. While the northeast region showed the highest contraceptive usage post abortion, south region had the lowest post-abortion contraceptive use. The use of SARC and traditional methods of contraception post-abortion was lowest in the south, and highest in the northeast region. Sterilizations were least prevalent post-abortions in the northeast region. These findings suggest that the choice of contraceptive methods is region-specific and thus call for region-specific interventions to address barriers and promote contraceptive uptake in different regions.

Although approximately 70% of the fieldwork in NFHS-5 was completed before the COVID-19 pandemic hit India, with the rest completed after the COVID-19 related lockdowns were lifted, there was no significant association between COVID-19 related lockdowns and post-abortion contraceptive use. This suggests that, while the COVID-19 pandemic did disrupt contraceptive service provision [49], its influence on post-abortion contraceptive use may have been less impactful. This may also reflect the unique method mix of post-abortion contraceptive users, relative to women of reproductive age in India, which more heavily relies on methods such as condoms, rhythm and withdrawal that do not require a health facility.

WHO guidelines suggest that women who have undergone abortions should be offered contraceptive counselling and services [1,50]. In our study, 23% of women accessed abortion outside a health facility (22% at home and 1% elsewhere). These women have limited options for receiving post-abortion contraceptive counselling and services unless they experience post-abortion complications that requires consultation with health providers. Moreover, there appears to be a lack of quality post-abortion contraceptive counselling and services in India as only one-fifth and one-third of women initiated contraceptive use within one month and three months of abortion, respectively. Studies also show more limited contraceptive counselling in private health facilities compared to public health facilities [51]. This is important given more than half of the abortions in our study happened in private health facilities. A recent Indian study showed that high-quality family planning counselling is associated with clients subsequently selecting more effective contraceptives, including IUD and sterilization [52]. These findings also highlight the opportunity for post-abortion contraceptive outreach, given that more than 30% of women who reported an abortion initiated contraception more than one month after their abortion.

A strength of this study is the use of most recent reproductive calendar data from a nationally representative survey from a large and diverse country like India. Second, large sample size enabled us to estimate multinomial probit

regression with greater confidence. Third, we adjusted our estimates for the reporting bias using Inverse Mills Ratio, which has often been ignored by the existing studies. Fourth, we adjusted our analysis for the COVID-19 related lockdowns in India. Fifth, unlike the previous studies on post-abortion contraception in the Indian context, we separated LARC, SARC, traditional methods, and sterilization in our analysis. LARCs are considered highly effective and safe for women of reproductive age, regardless of age, with few exceptions [53,54]. Our study is not without limitations. The reliance on self-reported data spanning over last 5 years may introduce recall bias, potentially affecting the accuracy of past-abortion contraceptive use, though a recent study from urban Kenya revealed little difference between shorter calendar (three years instead of five years) versus longer calendar [55]. The study lacks insights into the decision-making process regarding contraceptive use post-abortion, as well as women's contraceptive preferences relative to method availability. Additionally, important details like post-abortion counselling and services, intentions for contraception use post-abortion, and the role of husbands in post-abortion contraceptive decisions were not included in NFHS-5. Another limitation worth noting is that our study entirely measures initiation of post-abortion contraception. Women may discontinue or switch immediately, or after a few months. Initiation is important, but sustained use of contraceptive methods is something worth exploring in future research.

These findings have significant implications for reproductive healthcare programs and policies in India. It is crucial to integrate family planning services within abortion care, ensuring comprehensive information and counselling on contraceptive options during the post-abortion period. A study from Nigeria and Côte d'Ivoire showed that talking to someone about contraceptive use after abortion is a key factor influencing post-abortion contraceptive use [56]. As a majority of abortions in India happened in private health facilities, there is an urgent need to strengthen the quality of contraceptive counselling in private health facilities in India. Additionally, the growth in post abortion contraception points to an opportunity to expand access to contraception by making abortion more easily accessible and available in the public system – including by known strategies such as increasing trained providers, reducing legal impediments, expanding choice of methods and privacy accompanied by respectful and quality contraceptive counselling to support women to choose a method that is best for their own circumstances and preferences, delinking training on surgical abortion and medical abortion, etc. Given that about a quarter of women self-managed their abortion, there is an opportunity to improve messaging by pharmacists on post-abortion contraception. Expanding access to a broader range of contraceptive options and improving contraceptive counselling services should be a policy priority. Strengthening partnerships between private providers and public health programs can also help bridge this gap. A comprehensive abortion care policy is an important option that India can pursue in this direction. Policies directly addressing traditionally held social and cultural norms, such as son preference, are also needed to increase post-abortion contraceptive use in India. While policy intervention alone is inadequate to shift deeply entrenched norms, it offers a critical means of creating an enabling environment and offering more opportunities to promote equity and the status of women and girls. Community-based education and awareness programs, engagement of local influencers, and sensitization of healthcare providers may also be important avenues with which to tackle social stigma, lack of awareness, misconceptions, and cultural norms.

## Author contributions

**Conceptualization:** Joemet Jose, Ajit Kumar Kannaujiya.

**Data curation:** Joemet Jose, Ajit Kumar Kannaujiya.

**Formal analysis:** Joemet Jose, Ajit Kumar Kannaujiya.

**Supervision:** Kaushalendra Kumar, Abhishek Singh.

**Writing – original draft:** Kaushalendra Kumar, Abhishek Singh.

**Writing – review & editing:** Lotus McDougal, Katherine Hay.

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
