## [Decision Letter · Decision Letter 0]

18 Mar 2025

PONE-D-24-11169Post-abortion contraceptive use among currently married women in India: new evidence from National Family Health Survey 2019-2021 (NFHS-5)PLOS ONE

Dear Dr. Abhishek Singh,

Thank you for submitting your manuscript to PLOS ONE. After careful consideration, we feel that it has merit but does not fully meet PLOS ONE’s publication criteria as it currently stands. Therefore, we invite you to submit a revised version of the manuscript that addresses the points raised during the review process.

We look forward to receiving your revised manuscript.

Kind regards,

Jayanta Kumar Bora, PhD

Academic Editor

PLOS ONE

Journal Requirements:

“This work was supported by the Bill and Melinda Gates Foundation under grant numbers INV-008648 and INV-047355.”

3. Thank you for uploading your study's underlying data set. Unfortunately, the repository you have noted in your Data Availability statement does not qualify as an acceptable data repository according to PLOS's standards.

Reviewers' comments:

Reviewer's Responses to Questions

**Comments to the Author**

1. Is the manuscript technically sound, and do the data support the conclusions?

Reviewer #1: Yes

Reviewer #2: Yes

2. Has the statistical analysis been performed appropriately and rigorously? 

Reviewer #1: Yes

Reviewer #2: Yes

3. Have the authors made all data underlying the findings in their manuscript fully available?

Reviewer #1: Yes

Reviewer #2: Yes

4. Is the manuscript presented in an intelligible fashion and written in standard English?

Reviewer #1: Yes

Reviewer #2: Yes

5. Review Comments to the Author

Reviewer #1: Overall, this study makes a valuable contribution to understanding post-abortion contraceptive use in India. It is highly relevant for policy-making but would benefit from the below suggested improvements.

Comments for the Author

1. Abstract: If space permits, including a brief mention of the sample size and statistical methods used in the abstract would enhance the clarity for a broader audience.

2. Some abbreviations like LARC and SARC are not clearly defined when first introduced. Clarifying these would improve accessibility for the reader.

3. The article is well-written but some sections could be trimmed, especially repeated information on the importance of post-abortion contraceptive use.

4. In the Indian context, abortion is a highly sensitive issue, and women may be reluctant to report such incidents. This could result in underreporting of abortions in the reproductive calendar data which may result in biased association between factors affecting post abortion contraceptive use. How have you tackled this problem.

5. While the study is targeted at a specialist audience, a sentence or two simplifying complex terms would make it more accessible. For example terms like "Inverse Mills Ratio" (IMR) may not be familiar to all readers. Consider a brief explanation on the terms used wherever applicable.

6. It would help to briefly explain what is reproductive calendar data and how is it used in the analysis of this manuscript for those who are unfamiliar with it.

7. Attention to grammar, phrasing, and consistency in terminology will also contribute to a more polished and professional presentation of the research work. There are a few minor grammatical issues and typographical errors in the manuscript. Examples.

§ "These large increase in post-abortion contraceptive use in India since 2005-06..." can be corrected to "This large increase in post-abortion contraceptive use in India since 2005-06..."

§ "with the rest was completed after the COVID-19 related lockdowns were lifted..."can be corrected to "with the rest completed after the COVID-19 related lockdowns were lifted..."

§ "In 2019-21, 13% of women using PPIUCD received compensation." Can be corrected to "In 2019-21, 13% of women who used PPIUCD received compensation."

§ "The manuscript focuses on the important role media plays in improving contraceptive use." can be corrected to "The manuscript focuses on the important role that media plays in improving contraceptive use."

§ "Despite these efforts, the overall gain in post abortion family planning and the mix of methods promoted through these guidelines is not reflected..." can be corrected to "Despite these efforts, the overall gain in post-abortion family planning and the mix of methods promoted through these guidelines are not reflected...".

Reviewer #2: The study highlights the importance of integrating family planning into abortion care.The use of multinomial probit regression analysis is appropriate for examining factors associated with contraceptive use. A brief mention of specific policy recommendations could make the findings more actionable.

6. PLOS authors have the option to publish the peer review history of their article (what does this mean? ). If published, this will include your full peer review and any attached files.

**Do you want your identity to be public for this peer review?** For information about this choice, including consent withdrawal, please see our Privacy Policy .

Reviewer #1: **Yes: ** Dr. Illias Sheikh

Reviewer #2: No

---

## [Author Response · Author response to Decision Letter 1]

25 Mar 2025

Dated: March 24, 2025

To

Dr. Jayanta Kumar Bora

Academic Editor

PLOS ONE

Sub: Point-by-point reply to reviewer’s comments

Dear Dr. Bora

Thank you so much for considering our paper for possible publication in PLOS ONE. We are grateful to you for sharing the comments of reviewer with us. We have made maximum use of the comments to improve our paper. Please find below our point-by-point reply to the reviewer’s comments. I hope that you will find the revised paper suitable for publication in PLOS ONE.

We will be happy to provide more information if needed.

Yours Sincerely

Authors

Response to reviewer 1

Comment 1: Abstract: If space permits, including a brief mention of the sample size and statistical methods used in the abstract would enhance the clarity for a broader audience.

Response: Thank you for your valuable suggestion. We have revised the abstract to include details on the sample size and statistical methods used in our study. Specifically, we now mention that our analyses are based on a weighted sample of 5,103 women from NFHS-5 and 5,473 women from NFHS-4. Additionally, we have specified that a two-stage estimation procedure using the Inverse Mills Ratio (IMR) framework was employed to address potential biases in abortion reporting. In the second stage, a multinomial probit regression model was used to examine factors associated with post-abortion contraceptive use.

Comment 2: Some abbreviations like LARC and SARC are not clearly defined when first introduced. Clarifying these would improve accessibility for the reader.

Response: Thank you for your helpful suggestion. We have revised the manuscript to ensure that abbreviations such as LARC (long-acting reversible contraception) and SARC (short-acting reversible contraception) are clearly defined when first introduced. This can be seen under the Outcome Variable section in the Methods. We now define LARC as long-term, easily reversible contraceptive methods and SARC as methods that require regular use or application and are easily reversible.

Comment 3: The article is well-written but some sections could be trimmed, especially repeated information on the importance of post-abortion contraceptive use.

Response: Thank you for your insightful suggestion. We have carefully revised the manuscript to remove any redundant information, particularly in sections discussing the importance of post-abortion contraceptive use.

Comment 4: In the Indian context, abortion is a highly sensitive issue, and women may be reluctant to report such incidents. This could result in underreporting of abortions in the reproductive calendar data which may result in biased association between factors affecting post abortion contraceptive use. How have you tackled this problem?

Response: Thank you for your thoughtful comment. We agree that abortion is a very sensitive issue in India, leading to potential underreporting in surveys. To address this, we have adopted a two-stage estimation procedure using the Inverse Mills Ratio (IMR) framework proposed by James Heckman. In the first stage, we used a probit regression model to estimate the likelihood of abortion, incorporating several explanatory variables such as previous pregnancy outcomes, desired fertility, age at marriage, education, household type, wealth quintiles, caste, residence, and religion. The IMR derived from this model captures potential selection bias in abortion reporting. Following this, we included the IMR as an independent variable in our multinomial probit regression model, which examines factors associated with post-abortion contraceptive use. This helps correct the potential biases from underreporting of abortions and ensures more reliable estimates. We have mentioned this in detail under the statistical analysis section of the manuscript.

Comment 5: While the study is targeted at a specialist audience, a sentence or two simplifying complex terms would make it more accessible. For example, terms like "Inverse Mills Ratio" (IMR) may not be familiar to all readers. Consider a brief explanation on the terms used wherever applicable.

Response: Thank you for your suggestion. We have now included a brief explanation of IMR in the Statistical analysis section. Specifically, we clarify that IMR is a correction factor used in regression models to account for selection bias, ensuring more reliable estimates when certain outcomes, such as abortion reporting, may be systematically underreported.

Comment 6: It would help to briefly explain what is reproductive calendar data and how is it used in the analysis of this manuscript for those who are unfamiliar with it.

Response: We have already defined reproductive calendar data in the manuscript. However, to improve clarity, we have added additional information and repositioned it earlier in the text.

Comment 7: Attention to grammar, phrasing, and consistency in terminology will also contribute to a more polished and professional presentation of the research work. There are a few minor grammatical issues and typographical errors in the manuscript. Examples.

§ "These large increase in post-abortion contraceptive use in India since 2005-06..." can be corrected to "This large increase in post-abortion contraceptive use in India since 2005-06..."

§ "with the rest was completed after the COVID-19 related lockdowns were lifted..."can be corrected to "with the rest completed after the COVID-19 related lockdowns were lifted..."

§ "In 2019-21, 13% of women using PPIUCD received compensation." Can be corrected to "In 2019-21, 13% of women who used PPIUCD received compensation."

§ "The manuscript focuses on the important role media plays in improving contraceptive use." can be corrected to "The manuscript focuses on the important role that media plays in improving contraceptive use.”

§ "Despite these efforts, the overall gain in post abortion family planning and the mix of methods promoted through these guidelines is not reflected..." can be corrected to "Despite these efforts, the overall gain in post-abortion family planning and the mix of methods promoted through these guidelines are not reflected...".

Response: Thank you for your detailed feedback. We have carefully reviewed the manuscript and made the necessary grammatical and typographical corrections to improve clarity, and consistency. We have revised the sentences you highlighted to ensure proper phrasing and grammatical accuracy. Additionally, we have conducted a thorough proofreading of the entire manuscript to address any other minor inconsistencies.

Response to reviewer 2

Comment: The study highlights the importance of integrating family planning into abortion care. The use of multinomial probit regression analysis is appropriate for examining factors associated with contraceptive use. A brief mention of specific policy recommendations could make the findings more actionable.

Response: Thank you for your valuable feedback. We have incorporated specific policy recommendations to make the findings more actionable. These include strengthening contraceptive counselling and expanding the availability of a wider range of contraceptive methods.

---

## [Editor Report · Decision Letter 1]

28 Mar 2025

Post-abortion contraceptive use among currently married women in India: new evidence from National Family Health Survey 2019-2021 (NFHS-5)

PONE-D-24-11169R1

Dear Dr. Abhishek Singh,

We’re pleased to inform you that your manuscript has been judged scientifically suitable for publication and will be formally accepted for publication once it meets all outstanding technical requirements.

Kind regards,

Dr. Jayanta Kumar Bora

Academic Editor

PLOS ONE
---

## [Editor Report · Acceptance letter]

PONE-D-24-11169R1

PLOS ONE

Dear Dr. Singh,

I'm pleased to inform you that your manuscript has been deemed suitable for publication in PLOS ONE. Congratulations! Your manuscript is now being handed over to our production team.

Kind regards,

on behalf of

Dr. Jayanta Kumar Bora

Academic Editor

PLOS ONE